# The miRNA Profile in Non-Hodgkin’s Lymphoma Patients with Secondary Myelodysplasia

**DOI:** 10.3390/cells9102318

**Published:** 2020-10-19

**Authors:** Yuliya Andreevna Veryaskina, Sergei Evgenievich Titov, Igor Borisovich Kovynev, Tatiana Ivanovna Pospelova, Igor Fyodorovich Zhimulev

**Affiliations:** 1Laboratory of Gene Engineering, Institute of Cytology and Genetics, SB RAS, Novosibirsk 630090, Russia; 2Department of the Structure and Function of Chromosomes, Laboratory of Molecular Genetics Institute of Molecular and Cellular Biology, SB RAS, Novosibirsk 630090, Russia; titovse78@gmail.com (S.E.T.); zhimulev@mcb.nsc.ru (I.F.Z.); 3AO Vector-Best, Novosibirsk 630117, Russia; 4Department of Therapy, Hematology and Transfusiology, Novosibirsk State Medical University, Novosibirsk 630091, Russia; kovin_gem@mail.ru (I.B.K.); depart04@mail.ru (T.I.P.)

**Keywords:** myelodysplastic syndromes, myelodysplasia, miRNA, anemia, non-Hodgkin’s lymphoma

## Abstract

Myelodysplastic syndromes are a group of clonal diseases of hematopoietic stem cells and are characterized by multilineage dysplasia, ineffective hematopoiesis, peripheral blood cytopenias, genetic instability and a risk of transformation to acute myeloid leukemia. Some patients with non-Hodgkin lymphomas (NHLs) may have developed secondary myelodysplasia before therapy. Bone marrow (BM) hematopoiesis is regulated by a spectrum of epigenetic factors, among which microRNAs (miRNAs) are special. The aim of this work is to profile miRNA expression in BM cells in untreated NHL patients with secondary myelodysplasia. A comparative analysis of miRNA expression levels between the NHL and non-cancer blood disorders samples revealed that let-7a-5p was upregulated, and miR-26a-5p, miR-199b-5p, miR-145-5p and miR-150-5p were downregulated in NHL with myelodysplasia (*p* < 0.05). We for the first time developed a profile of miRNA expression in BM samples in untreated NHL patients with secondary myelodysplasia. It can be assumed that the differential diagnosis for blood cancers and secondary BM conditions based on miRNA expression profiles will improve the accuracy and relevance of the early diagnosis of cancerous and precancerous lesions in BM.

## 1. Introduction

Myelodysplastic syndromes (MDSs) are a group of clonal diseases of hematopoietic stem cells and are characterized by multilineage dysplasia in immature myeloid cells, ineffective hematopoiesis, peripheral blood (PB) cytopenias and a risk of transformation to acute myeloid leukemia (AML) [1]. According to the Surveillance, Epidemiology and End Results (SEER), the MDS incidence in the USA in 2012–2017 was 28,032 and the majority of the affected people were above 70 years of age [2]. Diagnosing MDS is challenging; for example, at early stages, it is quite difficult to differentiate minor morphological changes associated with dysplasia from other bone marrow (BM) deficiencies [3]. Not only can MDS be de novo, but also secondary, which cancer patients develop following treatment with cytostatic agents (t-MDS) [4]. Secondary MDS is associated with symptomatic PB cytopenias and abnormal BM cell morphology resembling those observed in primary MDS. One important discovery was the fact that patients with non-Hodgkin’s lymphomas (NHLs) can develop myelodysplasia even before the onset of treatment [5]. Additionally, one of the common concomitant pathologies in primary and secondary myelodysplasia is anemia. There are multiple factors responsible for anemia in lymphoma patients, including the anemia of chronic disease, iron deficiency anemia, autoimmune hemolytic anemia, marrow infiltration by tumor cells and blood loss. More than one of these factors may occur and be responsible for anemia even in a single patient [6].

Some researchers opine that NHL and myelodysplasia are not connected with each other and that the simultaneous occurrence of these pathologies is accidental; however, there now are three hypothesis seeking to explain this simultaneous occurrence: (1) pluripotent stem cells being a common origin, (2) patients’ specific immune status and (3) an increased level of cytokines and endothelial vascular growth factor (VEGF). However, the matter is still an issue [7,8].

Every stage of hematopoiesis is regulated by genetic and epigenetic factors, including miRNAs [9]. MicroRNAs (miRNAs) are short non-coding RNAs, which regulate gene expression. To date, more than 2600 human miRNAs have been identified, each having the potential to regulate hundreds of target genes [10]. MicroRNAs are important regulators of the differentiation and development of HSCs, and so their aberrant expression promote the development of both myeloid and lymphoid neoplasms [9]. The number of many hematopoiesis-associated miRNAs was substantially increased in the extracellular vesicles in MDS plasma [11]. In turn, miRNAs can be transported by exosomes and influence gene expression in BM cells. Thus, at least one more hypothesis can be proposed to explain NHL with secondary myelodysplasia: BM hematopoiesis can be affected by the microenvironment of a local tumor due to the variability of the miRNA content.

The profile of miRNA expression in untreated NHL patients with secondary myelodysplasia, whether with or without anemia, has for the first time been developed in this work. It can be assumed that the differential diagnosis for blood cancers and secondary BM conditions based on miRNA expression profiles will improve the accuracy and relevance of the early diagnosis of cancerous and precancerous lesions in BM.

## 2. Materials and Methods

### 2.1. Clinical Samples

This study was approved by the Ethics Committee of the Novosibirsk State Medical University (Novosibirsk, Russia). We used 118 cytological specimens obtained by a BM aspiration in the Municipal Hematological Center of the Ministry of Health of the Novosibirsk Region. The study groups were patients with MDS (*n* = 19), NHL without myelodysplasia (NHL (−MD); *n* = 22), NHL with myelodysplasia and without anemia (NHL (+MD) (−A) (*n* = 11)), NHL with myelodysplasia and anemia (NHL (+MD) (+A) (*n* = 8)); the control group consisted of patients with non-cancerous blood diseases (NCBD; *n* = 58)). The detailed clinical information of patients is listed in (Appendix A). All biological material was obtained in compliance with the legislation of the Russian Federation, written informed consent was provided by all patients, all data were depersonalized.

### 2.2. Isolation of Total RNA

Total RNA was extracted using the RNeasy Mini kit (QIAGEN, Valencia, CA, USA) according to the manufacture’s recommendations.

### 2.3. NanoString nCounter miRNA Assay for miRNA Profiling

The expression of 798 miRNAs was evaluated using the nCounter Human v3 miRNA Expression Assay kit (NanoString Technologies, Inc., Seattle, WA, USA) in accordance with the manufacturer’s protocol. For the NanoString assay, 100 ng of total RNA isolated from BM aspiration material. The RNA concentration was measured by a NanoDrop 2000 spectrophotometer (Thermo Fisher Scientific, Inc., Waltham, MA, USA). According to NanoString recommendations, the 260/280 ratio should be not less than 1.9 and the 260/230 ratio, not less than 1.8, and so were our figures. The data were analyzed using the nSolver v4 package (NanoString Technologies, Inc., Seattle, WA, USA). The nCounter assay for each sample consisted of six positive controls, eight negative controls and five control mRNAs (ACTB, B2M, GAPDH, RPL19 and RPLP0). Each sample was normalized to the geometric mean of the top 100 most highly expressed miRNAs. MiRNAs were selected as candidates if their differential expression in the comparison groups showed more than a 3-fold difference.

### 2.4. Validation of NanoString Results by RT-qPCR Analysis

miR-1246, miR-145-5p, miR-150-5p, miR-155-5p, miR-181a-5p, miR-199b-5p, miR-96-5p, let-7a-5p, miR-451a, miR-126-3p, miR-16-5p, miR-185-5p and miR-26a-5p were selected to validate their expression levels by RT-qPCR; the sequences of all oligos are listed in (Appendix A). The geometric mean of miR-378 and miR-191 expression levels was used for normalization. Analysis of NanoString nCounter miRNA quantification data showed that these miRNAs are present in the samples in numbers sufficiently above background numbers, but have the least variability between the comparison groups. Oligonucleotides were selected using the PrimerQuest online service (http://eu.idtdna.com/). Analysis of the threshold cycles generated by the qPCR was performed using the 2-ΔCt method [12]. Statistical analysis was performed using Statistica v13.1. The Mann–Whitney U test was used to examine the statistical difference in clinical parameters between samples derived from the comparison groups. *p*-values < 0.05 indicate statistical significance.

### 2.5. Reverse Transcription

The reverse transcription reaction for cDNA was carried out in a volume of 30 µL. Ready-for-use reactions RealBest RT Master Mix (Vector-Best, Novosibirsk, Russia) were utilized. The reverse transcription reaction contained 3 μL of RNA preparation, 21.6% trehalose, 1× RT buffer (Vector-Best, Novosibirsk, Russia), 0.4 mM of each dNTP, 1% BSA, 100U M-MLV reverse transcriptase (Vector-Best, Novosibirsk, Russia) and 0.2 μM of appropriate RT primer. All oligonucleotides were synthesized by Vector-Best (Novosibirsk, Russia). Three microliters of the reaction mixture containing cDNA was used immediately as a template for a real-time PCR on a CFX96 system (Bio-Rad, Hercules, CA, USA).

### 2.6. Real-Time PCR

MiRNA expression levels were measured by real-time PCR on a CFX96 amplifier (Bio-Rad Laboratories, Hercules, CA, USA). The total volume of each reaction was 30 μL and contained 3 μL of cDNA, 1× PCR buffer (Vector-Best, Novosibirsk, Russia), 0.4 mM of each dNTP (Biosan, Riga, Latvia), 1% BSA, 1U Taq polymerase (Vector-Best, Novosibirsk, Russia) premixed with 10× active center-specific monoclonal antibody (Clontech, Mountain View, USA), 0.5 units of uracil-DNA glycosylase (Vector-Best, Novosibirsk, Russia), 0.5 μM of each primer and 0.25 μM of TaqMan probe. The primers and the probes are Vector-Best developments, the efficiency of the PCR being 90–100%.

### 2.7. Functional Analysis of miRNAs Using DIANA-miRPath

The search for experimentally supported miRNA targets was made using DIANA-TarBase v8. (Thessaly, Greece). The pathway analysis was carried out by DIANA-miRPath v3.0 (Thessaly, Greece) using *p* < 0.05 as a significant threshold.

## 3. Results

In this study, we analyzed the expression of 798 miRNA from NHL, MDS and NCBD, using the NanoString nCounter analysis system. To subtract the background noise, miRNAs with expression less than two standard deviations from the mean of negative controls were not included in the analysis. This threshold allowed us to obtain 84 miRNAs suitable for analysis. We performed a comparative analysis of the number of these miRNAs between MDS, NHL (−MD), NHL (+MD) (−A), NHL (+MD) (+A) and NCBD groups and identified 62 miRNAs, the number of which was different in the comparison groups (Appendix A). Next, we selected those miRNAs, the differential expression of which in the comparison groups showed more than a 5-fold difference. For further validation, we selected 13 miRNAs (miR-1246, miR-145-5p, miR-150-5p, miR-155-5p, miR-181a-5p, miR-199b-5p, miR-96-5p, let-7a-5p, miR-451a, miR-126-3p, miR-16-5p, miR-185-5p and miR-26a-5p) in the comparison groups.

### 3.1. The Profile of miRNA Expression in BM Tumor Samples and NCBD

The expression levels of miRNAs were measured by RT-qPCR in the MDS, NHL (−MD), NHL (+MD) (−A), NHL (+MD) (+A) and NCBD groups (Figure 1).

A comparative analysis of miRNA expression levels between the cancer and NCBD samples revealed that miR-26a-5p, miR-181a-5p, miR-185-5p, miR-96-5p, miR-1246, miR-199b-5p, miR-145-5p and miR-150-5p were downregulated in NHL (−MD; *p* < 0.05); let-7a-5p was upregulated and miR-26a-5p, miR-199b-5p, miR-145-5p and miR-150-5p were downregulated in NHL (+MD) (−A) (*p* < 0.05); let-7a-5p was upregulated and miR-26a-5p, miR-96-5p, miR-1246, miR-199b-5p, miR-145-5p and miR-150-5p were downregulated in NHL (+MD) (+A) (*p* < 0.05) and let-7a-5p was upregulated and miR-26a-5p, miR-181a-5p, miR-185-5p, miR-1246, miR-199b-5p, miR-145-5p and miR-150-5p were downregulated in MDS (*p* < 0.05; Table 1).

Evidence existed that the BM levels of miR-26a-5p, miR-1246, miR-199b-5p, miR-145-5p and miR-150-5p were substantially decreased in both lymphoid and myeloid tumors compared with NCBD samples (*p* < 0.05). The expression levels of miR-181a-5p and miR-185-5p decreased in NHL (−MD) and MDS (*p* < 0.05) compared to NCBD, but the differences between NHL with myelodysplasia and the NCBD samples failed to reach significance. We observe a tendency towards a decrease in the expression levels of miR-96-5p in NHL, no matter whether with or without myelodysplasia. It should be noted that the expression level of let-7a-5p was substantially increased in all myelodysplastic groups compared to the NCBD groups (*p* < 0.05).

We observe a tendency towards a decrease in the expression levels of miR-96-5p and miR-1246 in NHL samples with an anemic syndrome in combination with myelodysplasia (*p* < 0.05); however, for this observation to be confirmed, larger sample sizes are required.

### 3.2. Bioinformatic Analysis of Genes Targeted by miRNAs and the Pathways Involved in Cancer Processes

We found that let-7a-5p, miR-26a-5p, miR-181a-5p, miR-185-5p, miR-96-5p, miR-1246, miR-199b-5p and miR-145-5p were the most important miRNAs in NHL and MDS pathogenesis. DIANA-TarBase v8 experimentally confirmed miRNA targets to be identified. The criteria for a gene to be considered a miRNA target were as follows: an mRNA–miRNA interaction in BM tissue was proven experimentally and reported by at least two organizations; if it was reported by a single organization, the prediction score should be higher than 0.9 (Appendix A).

DIANA-TarBase v8 revealed 61 enriched pathways with adjusted *p* < 0.05 Next, we concentrated on 19 pathways immediately associated with cancer. We found that all miRNAs in question were involved in each of these pathways. The results are summarized in Table 2.

## 4. Discussion

Molecular genetic markers gradually become more and more popular in describing MDS and allow MDS subtypes to be discriminated from each other. MDS is accompanied by genetic and epigenetic changes, including aberrant expression of miRNAs [13]. Nevertheless, from a fundamental point of view, further research is needed to understand the complex regulatory mechanisms between miRNAs and their target genes in MDS. From a clinical point of view, it is important to differentiate MDS from other BM pathologies, including secondary myelodysplasias in NHL. In this work, we developed a profile of miRNA expression in (1) MDS patients and (2) NHL patients with and without secondary myelodysplasia against NCBD peers. Of most interest here is the absolutely first miRNA expression profile in NHL patients who had not been treated with cytostatic agents but developed myelodysplasia nonetheless. Thus, let-7a-5p was upregulated and miR-26a-5p, miR-199b-5p, miR-145-5p and miR-150-5p were downregulated in NHL (+MD) (−A) compared with NCBD (*p* < 0.05), and let-7a-5p was upregulated and miR-26a-5p, miR-96-5p, miR-1246, miR-199b-5p, miR-145-5p and miR-150-5p were downregulated in NHL (+MD) (+A) compared with NCBD (*p* < 0.05).

These miRNAs are among the regulators of normal hematopoiesis. Therefore, changes in their expression levels contribute to the development of hematologic neoplasms. Thus, high miR-150 expression during myelopoiesis promotes megakaryopoiesis, while low miR-150 expression leads to erythropoiesis [14]. Moreover, decreases in miR-150 expression in blood cancers such as acute lymphoblastic leukemia, AML, mantle cell lymphoma, MDS and diffuse large B cell lymphoma (DLBCL) had previously been reported [14,15,16,17].

Another important player in normal hematopoiesis is miR-145. Interestingly, miR-146a and miR-145 are involved in megakaryopoiesis by activating the innate immunity targets TIRAP and TRAF6 [18]. Experiments with mice showed that the absence of miR-145 contributes to a decrease in HSCs followed by a preleukemic state, anemia and thrombocytopenia [19]. As pointed out in another work, miR-145 and miR-146a knockdown promotes thrombocytosis, mild neutropenia and megakaryocytic dysplasia [20]. It is noted that a decrease in miR-145 expression promotes the progression of MDS and diffuse large B cell lymphomas [21,22].

A large number of works mentioning the role of miR-26a as a suppressor of various cancer types have been published [23,24,25,26,27]. In particular, the expression level of miR-26a-5p is substantially decreased in AML and DLBCL [28,29].

The preservation of the HSC population is critical for sustained hematopoiesis in adults. Karaczyn et al. showed that decreased miR-199b-5p expression leads to an imbalance between long-term hematopoietic stem cells, short-term hematopoietic stem cells and multipotent progenitors [30]. Additionally, Li et al. point out that miR-199b-5p participates in the regulation of erythropoiesis [31].

Houshmand et al. demonstrated that miR-1246 expression is typically increased during megakaryocyte differentiation [32]. Additionally, miRNA-1246 regulates the expression of EBF1, which promotes the development and proliferation of B cells through activation of the AKT signaling pathway [33].

We found that miRNA-96-5p expression was substantially decreased in NHL, no matter with or without myelodysplasia; however, statistically significant data were related only to NHL (MD) (−A) and NHL (+MD) (+A) (*p* < 0.05). Azzouzi et al. demonstrated that miRNA-96 directly suppresses γ-globin expression [34]. It can be assumed that the aberrant expression of miRNA-96 is part of the molecular genetic changes underlying anemia.

In this work, we found that the expression level of let-7a-5p is substantially increased in all myelodysplastic groups, NHL (+MD) (−A), NHL (+MD) (+A) and MDS, compared with NCBD (*p* < 0.05). Lessard et al. demonstrated that let-7a participates in the regulation of erythropoiesis [35]. In this work, we found a statistically significant difference in let-7a expression between NHL (+MD) (+A) and NCBD (*p* < 0.05). In this context, let-7a may be one of the regulators of impaired erythropoiesis and its aberrant expression can be associated with anemia.

A number of studies have been published analyzing miRNA expression levels in MDS and indicating decreased let-7a [36,37]. These data are inconsistent with our findings; however, the dynamics of changes in miRNA expression levels often varies even in the same disease, depending on the type of material being studied and the methods of analysis and data interpretation. While Zuo et al. showed that let-7a expression is decreased in MDS patients’ plasma, Hustincova et al. reported the reverse [11,38]. The inconsistency of data may be due to various technical reasons relating to the methods of analysis and mathematical interpretation of the data obtained.

It is expected that each miRNA regulates more than a hundred genes [39]. Various algorithms have been developed for identifying miRNA targets, but only some of these have been confirmed experimentally. Bioinformatic analysis showed that let-7a-5p, miR-26a-5p, miR-181a-5p, miR-185-5p, miR-96-5p, miR-1246, miR-199b-5p and miR-145-5p target genes that regulate cell proliferation, cell migration, cell cycle control, apoptosis and angiogenesis.

We found associations between the misregulation of the miRNAs and cellular processes in cancer using DIANA miRPath v.3.0 [40]. Overall, this analysis has revealed 61 regulatory pathways, of which we selected 19 associated with cancer. These results suggest that the misregulation of these miRNAs can affect the cellular processes associated with cancer, contributing to neoplasm initiation, development and progression [41,42,43,44].

In summary, an important task set before clinical oncology is the search for additional molecular-genetic markers with a possibility of integrating them into the existing international prognostic systems, and some of the most promising markers in this respect are miRNAs. Current data on the roles of miRNAs in myelodysplasias suggest that these molecules have the potential to be used as tools for the diagnosis and prognosis of MDS and may have relevance to the response to treatment. The prognostic significance of miRNA requires further validation in a larger patient cohort. Thus, the search for additional prognostic markers for the diagnosis of both de novo MDS and secondary lesions to BM will allow treatment to be personalized in the most precise manner.

## Figures and Tables

**Figure 1 cells-09-02318-f001:**
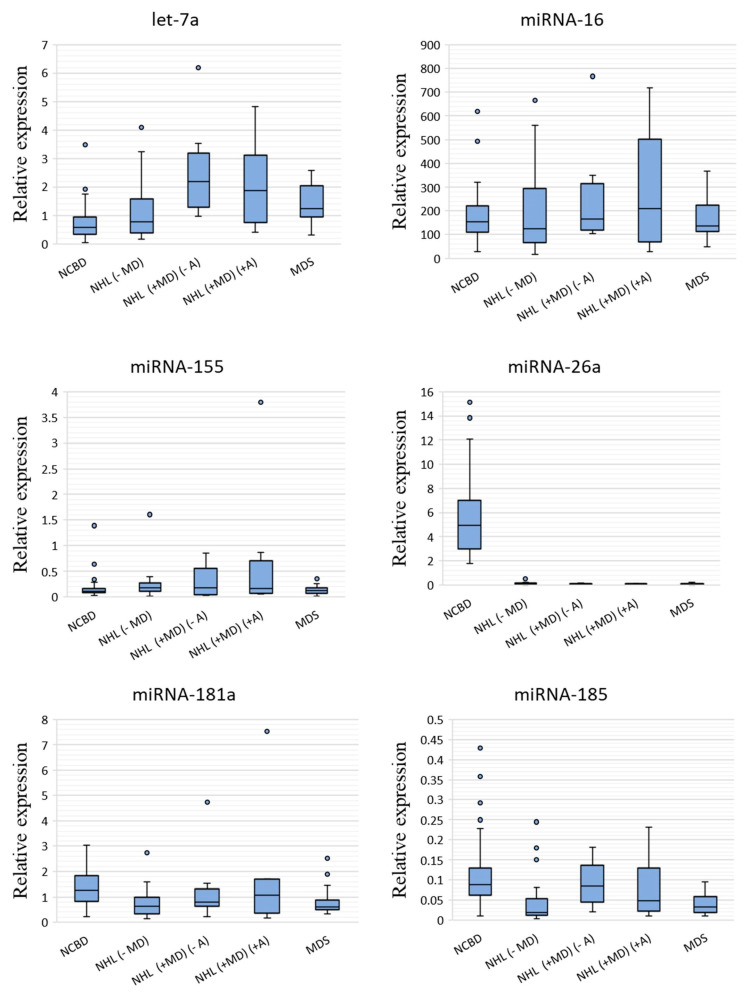
miRNA relative expression levels in non-Hodgkin’s lymphomas without myelodysplasia (NHL (−MD); *n* = 22), non-Hodgkin’s lymphomas with myelodysplasia and without anemia (NHL (+MD) (−A); *n* = 11), non-Hodgkin’s lymphomas with myelodysplasia and anemia (NHL (+MD) (+A); *n* = 8) and myelodysplastic syndrome (MDS; *n* = 19) against non-cancerous blood diseases (NCBD; *n* = 58). The figure presents the median value, the upper and lower quartiles, the non-outlier range and outliers appearing as the circles.

**Table 1 cells-09-02318-t001:** Comparative analysis of miRNA expression levels between tumor samples and NCBD.

	NHL (−MD) vs. NCBD	*p*-Value	NHL (+MD) (−A) vs. NCBD	*p*-Value	NHL (+MD) (+A) vs. NCBD	*p*-Value	MDS vs. NCBD	*p*-Value
let-7a-5p	1.35	0.122208	3.78	**0.000002**	3.25	**0.002901**	2.14	**0.000021**
miR-16-5p	−1.23	0.647554	1.08	0.333132	1.37	0.555872	−1.12	0.985981
miR-155-5p	1.53	0.053346	1.54	0.739425	1.44	0.444910	1.04	0.783056
miR-26a-5p	−51.97	**7.4 × 10^−18^**	−54.83	**3.83 × 10^−11^**	−44.82	**1.4 × 10^−9^**	−50.16	**8.2 × 10^−17^**
miR-181a-5p	−1.98	**0.000162**	−1.58	0.131210	−1.18	0.388964	−2.13	**0.000706**
miR-185-5p	−4.50	**0.000003**	−1.04	0.567289	−1.79	0.097465	−2.67	**0.000002**
miR-96-5p	−4.43	**0.000001**	−2.48	0.067222	−2.73	**0.003131**	−1.77	0.102073
miR-1246	−2.38	**0.000421**	−2.18	0.175334	−1.48	**0.029906**	−3.48	**0.007841**
miR-199b-5p	−1.98	**0.004179**	−3.39	**0.000225**	−2.59	**0.000908**	−1.79	**0.000355**
miR-126-3p	−1.06	0.497818	−1.16	0.884774	−1.10	0.839624	−1.25	0.237218
miR-451a	1.29	0.720385	1.01	0.739425	−2.22	0.239566	−1.54	0.065864
miR-145-5p	−8.31	**0.000000**	−5.05	**0.000002**	−2.27	**0.008440**	−4.35	**0.000000**
miR-150-5p	−1.66	**0.007797**	−1.67	**0.000426**	−2.27	**0.024672**	−2.54	**0.002474**

(NHL without myelodysplasia (NHL (−MD)), NHL with myelodysplasia and without anemia (NHL (+MD) (−A)), NHL with myelodysplasia and anemia (NHL (+MD) (+A)), myelodysplastic syndrome (MDS) and non-cancerous blood diseases (NCBD), statistically significant differences are in bold (*p* < 0.05)).

**Table 2 cells-09-02318-t002:** Cancer-associated pathways in which the miRNAs in question are involved. The list was generated by DIANA-mirPath v3.0.

KEGG Pathway	Genes in the Pathway, Total	*p*-Value
**Signaling pathway**		
Hippo signaling pathway (hsa04390)	86	2.3 × 10^−14^
TGF-beta signaling pathway (hsa04350)	54	1.4 × 10^−11^
Signaling pathways regulating pluripotency of stem cells (hsa04550)	77	1.2 × 10^−8^
FoxO signaling pathway (hsa04068)	72	4.2 × 10^−5^
HIF-1 signaling pathway (hsa04066)	58	5.8 × 10^−5^
ErbB signaling pathway (hsa04012)	43	2 × 10^−3^
MAPK signaling pathway (hsa04010)	107	5 × 10^−3^
PI3K-Akt signaling pathway (hsa04151)	142	1 × 10^−2^
mTOR signaling pathway (hsa04150)	32	2 × 10^−2^
Wnt signaling pathway (hsa04310)	58	4 × 10^−2^
**Cell biology**		
Cell cycle (hsa04110)	75	3.5 × 10^−11^
Apoptosis (hsa04210)	38	4 × 10^−2^
RNA transport (hsa03013)	73	2 × 10^−2^
RNA degradation (hsa03018)	39	3 × 10^−2^
**Cancer-associated pathways**		
Transcriptional misregulation in cancer (hsa05202)	85	1.1 × 10^−6^
Pathways in cancer (hsa05200)	175	3.6 × 10^−5^
Proteoglycans in cancer (hsa05205)	111	8.6 × 10^−16^
Chronic myeloid leukemia (hsa05220)	39	1 × 10^−3^
Bladder cancer (hsa05219)	25	2 × 10^−3^

(Kyoto Encyclopedia of Genes and Genomes (KEGG), transforming growth factor beta (TGF-beta), hypoxia-inducible factor 1-alpha (HIF-1), group of receptor tyrosine kinases (ErbB), the Mitogen-Activated Protein Kinase (MAPK), phosphatidylinositol 3-kinase (PI3K) and Akt/Protein Kinase B (PI3K-Akt), mammalian target of rapamycin (mTOR), group of signal transduction pathways (Wnt)).

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
