# Peer review of "The miRNA Profile in Non-Hodgkin’s Lymphoma Patients with Secondary Myelodysplasia"

_cells, 2020, doi:10.3390/cells9102318_

Round 1
Reviewer 1 Report
The authors question the miR expression profile of bone marrow cells from patients with non hodgkin lymphoma (NHL), patients with NHL and a secondary myelodysplastic syndrome (MDS) and patients with non cancerous blood disease. The rationale of this study is reinforced by the generic role of miR in MDS microenvironment remodeling, and long-distance action of miR transported by exosomes. The manuscript is clearly written. The design of the study is rather simple, with small cohorts but the differences observed are statistically significant. However, the work could have gained in depth if the consequences of the modulation of miR profiles would have been more investigated.
Major comments
In the abstract :
Line 44. The authors must elaborate more on the rationale of studying NHL and secondary MDS. In particular, we expect to find information on the incidence of those cases or the proportion of NHL patients developing MDS, before or after treatments. Moreover, is there some literature giving elements of comprehension at the molecular or physiology level to explain NHL associated MDS?
In the abstract :
The section on anemia is confusing. In particular because it seems that anemia is not viewed as a result of MDS. The idea of « concomitant » does not imply here a causal link. But in case of MDS, anemia is also the consequence of lack of effective erythropoiesis and, sometimes, also due to higher level of hepcidin that regulation plasma iron. Please, reformulate.
Lines 63-64 : the sequence is very unclear between NHL cells and MDS cells. If the authors postulate that BM hematopoiesis is affected by the microenvironment of a local tumor due to the variability of the miRNA content, what is the causal relationship with NHL ? Do the authors only consider in this study NHL originating from BM ? Do the authors hypothesize that NHL (even from non-BM tissue) impact on BM environment ? Please reformulate with more clarity.
- Indicate whether the BM cells have been sorted? Do the authors use mononuclear cells ? Do the samples also contain also BM micro-environment cells ?
129-139. Please provide in a supplemental data, the list of the 84 and 62 miR. Also explain the rationale for the selection of the final 13 miR.
Figure 1 : Statistical analysis results must appear in the graphs, at least by comparison to NCBD. Moreover, the Y-axes should read “relative expression (fold change)”. Table 1 will then be redundant and can be considered as a supplemental Table.
Table 2 is not an added-value to the manuscript. It corresponds to a list of putative MiR target with no biological validation in NHL. The authors must validate the most interesting targets by RT-PCR in NHL +/- MDS compared to NCBD.
Table 3 is poorly informative, just presented like this. The authors must select some genes of the most interesting pathways and validate their mysregulation in the different conditions (NHL +/- MDS compared to NCBD).
In the discussion, the authors must mention that a larger cohort will be required to validate their findings.
Minor
Line 17, line 35 : high risk of transformation> remove ‘high’. It really depends on the MDS subtype.
Line 22 : not clear ‘cancer and non-cancer blood disorders samples’. Do you mean MDS versus non MDS cells ? do you mean MDS in patients with NHL versus MDS in patients with non other cancer ? mean something else ?. Please clarify.
Line 24 : “… were downregulated in NHL with myelodysplasia” , please precise compared to what condition.
Line 37 : it is better to express the incidence as a proportion or a rate, not a raw number. (The total number of inhabitants in the USA is not obvious for everybody). Please modify.
Line 46 : “…responsible for anemia in patients with lymphoproliferative disorders”. In this manuscript, the authors are considering myeloproliferative and MDS, not lymphoproliferative. Please, edit, or make it clearer.
Line 61 : Many types of vesicles exist. Precise which one the authors are talking about.
Line 62 : “target cell of exosomes”. Again, precise or give examples. Here the purpose is to give the readers an idea of how distant can be those cells, in particular, are they cells within the BM or in different tissues.
Lines 146-151 : in the Figure 1 legend, indicate the number of samples per conditions (n=…). Indicate in the figure legend, what statistical analysis has been performed to compare the conditions. Indicate also what the relative expression (fold change) has been expressed, or put this in the material and methods.
Line 186. The full list of pathways must be presented as supplemental data. (see the major comment).
Line203. « and let-7a-5p was upregulated and miR-26a-5p, miR-96-5p, miR-1246, miR-199b-5p, miR-145-5p and miR-150-5p were downregulated in NHL (+MD) (+A) (p < 0.05). » compared to what condition(s) ?
Reviewer 2 Report
In this study, Veryaskina et al describe the miRNA profile of patients with NHL that develop secondary MDS and compare it with the profile of patients with non-hematological neoplasms, and isolated NHL and MDS. The paper is well-written, but the development of the results and discussion are difficult to follow for the reader because of the large amount of information offered by the authors. I only have minor comments aiming to improve the understanding of the study.
- NHL and MDS are heterogeneous group of diseases and the authors treat them as unique groups. The entities and number of cases included in each group should be described at least in a supplementary table. In addition, a rationale should be provided to study such categories instead of specific entities. The authors should also describe better the characteristics of the control group.
- Did the authors find differences in the miRNA profile among specific entities included in the whole categories?
- The results focus in the differences between the defined categories and control group. Did the author find relevant differences between the different categories defined?
- Discussion is too long and should be simplified. The large amount of information obtained is difficult to be easily report, but the authors should facilitate the understanding of their work to the reader.
Round 2
Reviewer 1 Report
The revised version has gained in clarity compared to the previous version.
This manuscript is a resubmission of an earlier submission. The following is a list of the peer review reports and author responses from that submission.